# A mating-induced reproductive gene promotes *Anopheles* tolerance to *Plasmodium falciparum* infection

Perrine Marcenac[1], W. Robert Shaw[1], Evdoxia G. Kakani[1¤a], Sara N. Mitchell[1¤a], Adam South[1¤b], Kristine Werling[1], Eryney Marrogi[1], Daniel G. Abernathy[1], Rakiswendé Serge Yerbanga[2], Roch K. Dabiré[2], Abdoulaye Diabaté[2], Thierry Lefèvre[2,3], Flaminia Catteruccia[1]*

1 Department of Immunology and Infectious Diseases, Harvard T. H. Chan School of Public Health, Boston, Massachusetts, United States of America, 2 Institut de Recherche en Sciences de la Santé/Centre Muraz, Bobo-Dioulasso, Burkina Faso, 3 MIVEGEC, IRD, CNRS, University of Montpellier, Montpellier, France

¤a Current address: Verily Life Sciences, South San Francisco, California, United States of America
¤b Current address: Department of Infectious Disease and Global Health, Cummings School of Veterinary Medicine at Tufts University, Grafton, Massachusetts, United States of America
* fcatter@hsph.harvard.edu

**Data Availability Statement:** All data and code files are available from the Harvard Dataverse repository: https://doi.org/10.7910/DVN/GU8DOA.

## Abstract

*Anopheles* mosquitoes have transmitted *Plasmodium* parasites for millions of years, yet it remains unclear whether they suffer fitness costs to infection. Here we report that the fecundity of virgin and mated females of two important vectors—*Anopheles gambiae* and *Anopheles stephensi*—is not affected by infection with *Plasmodium falciparum*, demonstrating that these human malaria parasites do not inflict this reproductive cost on their natural mosquito hosts. Additionally, parasite development is not impacted by mating status. However, in field studies using different *P. falciparum* isolates in *Anopheles coluzzii*, we find that *Mating-Induced Stimulator of Oogenesis* (*MISO*), a female reproductive gene strongly induced after mating by the sexual transfer of the steroid hormone 20-hydroxyecdysone (20E), protects females from incurring fecundity costs to infection. *MISO*-silenced females produce fewer eggs as they become increasingly infected with *P. falciparum*, while parasite development is not impacted by this gene silencing. Interestingly, previous work had shown that sexual transfer of 20E has specifically evolved in *Cellia* species of the *Anopheles* genus, driving the co-adaptation of MISO. Our data therefore suggest that evolution of male-female sexual interactions may have promoted *Anopheles* tolerance to *P. falciparum* infection in the *Cellia* subgenus, which comprises the most important malaria vectors.

## Author summary

*Plasmodium falciparum*, the deadliest form of human malaria, is transmitted when female *Anopheles* mosquitoes bite people and take a blood meal in order to develop eggs. To date, it is still poorly understood whether *Anopheles* mosquitoes that get infected with *P. falciparum* suffer fitness costs. Here, we find that the number of eggs produced by *Anopheles*

**Funding:** F.C. is funded by a Faculty Research Scholar Award by the Howard Hughes Medical Institute (HHMI) and the Bill & Melinda Gates Foundation (BMGF) (Grant ID: OPP1158190, www.hhmi.org), and by the National Institutes of Health (NIH) (R01 AI124165, R01 AI104956, www.nih.gov). P.M. was supported by the National Science Foundation Graduate Research Fellowship Program (Grant No. DGE1144152, www.nsf.gov), the Barry R. and Irene Tilenius Bloom Fellowship at the Harvard T. H. Chan School of Public Health (www.hsph.harvard.edu), and by a Merit/Term-time Research Fellowship from the Harvard University Graduate School of Arts and Sciences (www.gsas.harvard.edu). The funders had no role in study design, data collection and analysis, decision to publish, or preparation of the manuscript.

**Competing interests:** The authors have declared that no competing interests exist.

*gambiae* and *Anopheles stephensi* females is not affected by *P. falciparum* infection, and that the mating status of the mosquitoes does not impact the parasite. However, in field experiments infecting a related species, *Anopheles coluzzii*, with *P. falciparum* using blood from donors in Burkina Faso, we find that interfering with the expression of a gene normally triggered by the sexual transfer of the steroid hormone 20-hydroxyecdysone induces increasing costs to egg development as females become more infected with *P. falciparum*, with no impacts on the parasite. The results of our study suggest that pathways triggered by mating may help *Anopheles* prevent reproductive costs associated with *P. falciparum* infection, providing new insights into evolutionary strategies adopted by anophelines in the face of a longstanding association with *Plasmodium* parasites.

## Introduction

Parasites of the genus *Plasmodium* are the causative agents of malaria, a disease that infects hundreds of millions and kills over 400,000 people, mostly children, every year [1]. *Plasmodium* transmission occurs via the infectious bite of mosquitoes of the *Anopheles* genus, which comprises three major subgenera whose last common ancestor dates to 100 million years ago [2,3]: *Cellia* found in Africa, the Indian sub-continent, South East Asia, and Oceania; *Nyssorhynchus* located in Central and South America; and *Anopheles* distributed from North America to Asia [4]. *Cellia* species from the African continent have been vectors of *Plasmodium falciparum*, the deadliest malaria parasite, for the last 10,000–50,000 years [5–8], and have transmitted ancestral *Plasmodium* species for millions of years [9]. Given their long-term association with *P. falciparum* and its ancestral forms, an intriguing question is whether these vectors suffer fitness costs to infection.

Two major fitness costs possibly associated with infection are reduced longevity and impaired reproductive output. A meta-analysis on studies examining the effects of infection on mosquito survival found no costs to longevity when mosquitoes were infected with *P. falciparum*, while infections with rodent malaria parasites reduced female lifespan [10]. Moreover, rodent *Plasmodium* species also reduced the ability of females to develop eggs [11–16]. We recently began directly addressing whether *P. falciparum* induces reproductive costs in *Anopheles gambiae*, its predominant natural *Cellia* vector in sub-Saharan Africa [17]. Using virgin females, we unexpectedly found that egg numbers were positively correlated with *P. falciparum* infection intensity, so that females that produced more eggs in their ovaries harbored greater parasite numbers in their midgut. This correlation was shown to be mediated by 20-hydroxyecdysone (20E)—a steroid hormone produced by the female after blood feeding that plays an essential role in egg development [17–19]—as the positive link between parasite development and oogenesis was abolished by silencing either component of the 20E heterodimer nuclear receptor (ecdysone receptor EcR and ultraspiracle USP) [17]. Moreover, we and others have found that parasites can utilize lipids carried by the mosquito lipid transporter lipophorin for their own growth [17,20,21], but they do so without impairing egg development [17]. Although these results suggest a non-competitive *P. falciparum* developmental strategy tailored to *An. gambiae* reproductive biology [17], the study used virgin females and therefore did not determine the full costs potentially inflicted by parasites in reproductively active, mated females.

Mating profoundly reshapes female physiology, including initiating sperm storage and preservation in the spermatheca, a costly process that is essential for fertilization [22–24]. As a possible consequence of this increased investment in reproductive functions, mated

females may be less equipped to face *P. falciparum* infection, shifting the balance towards a competitive vector-parasite association. This issue is potentially exacerbated by the fact that female *An. gambiae* are monandrous, i.e., they mate a single time in their lives and therefore must preserve sperm viability over their entire lifespan [25,26]. Interestingly, monandry in this species is enforced by the same ecdysteroid hormone 20E, in this case produced by males in their accessory glands and transferred to the female atrium (uterus) during mating [27]. Besides enforcing monandry, male 20E transfer is responsible for an array of post-mating responses critical for female reproduction, including improved fertility, triggering of egg laying, and induction of large transcriptional changes in the female atrium [26–29]. Among the genes strongly upregulated by sexual transfer of 20E is the *Mating-Induced Stimulator of Oogenesis* (*MISO*), whose expression is specifically induced in the atrium where this female reproductive protein interacts directly or indirectly with 20E, favoring its function [28].

Interestingly, phylogenetically-based evolutionary analyses have revealed that sexual transfer of 20E is an acquired trait that has evolved specifically in *Cellia* [29] where levels of this male steroid hormone are highly variable between species, and it is largely absent in males from the *Nyssorhynchus* and *Anopheles* subgenera [29,30]. In turn, evolution of this male reproductive trait has induced reciprocal adaptations in female reproductive genes, including *MISO*, profoundly shaping female post-mating biology [29].

In an effort to better understand the interaction of *P. falciparum* parasites with their *Anopheles* vectors and gain insights into possible reproductive fitness costs associated with infection, we examined whether *P. falciparum* impairs an important component of female reproductive fitness—egg development after the first blood meal—in mated females using *An. gambiae* as well as *Anopheles stephensi*, a *Cellia* vector from the Indian sub-continent. We show that the fecundity of both anophelines is not impacted by infection regardless of mating status, and that *P. falciparum* infection intensity and prevalence are not affected by mating. However, in field studies using natural *P. falciparum* isolates, we uncover fitness costs to infection when we silence the 20E-responsive gene *MISO*, as MISO-depleted females produce fewer eggs when they become increasingly infected, while parasite numbers are unaffected. Therefore, this mating-induced gene supports the reproductive fitness of *P. falciparum*-infected *Anopheles* females not by limiting parasite infection, but rather by reducing fitness costs associated with it, an evolutionary strategy known as tolerance [31–33]. All together, these studies demonstrate that reproductively active females of *Cellia* species do not suffer costs to their fecundity when infected with *P. falciparum*, and they reveal a mating-induced factor critical to maintaining mosquito tolerance to *Plasmodium* infection.

## Results

### *An. gambiae* and *An. stephensi* females do not suffer fecundity costs to *P. falciparum* infection regardless of their mating status

We analyzed the effects of infection on egg development in both *An. gambiae* (G3 strain) and *An. stephensi* (SDA-500 strain) mated females. After mating, females from both species were fed on a blood meal containing *P. falciparum* (NF54) parasites or a heat-inactivated, non-infectious blood meal, using virgins as controls. Forty-eight hours (h) post-infectious blood feeding (pIBF), the number of eggs developed per mosquito was counted, while mated status was confirmed by detecting successful mating plug transfer prior to blood feeding (see Materials and Methods). In both species, virgin and mated females produced the same number of eggs whether the blood meal was infectious or non-infectious (**Fig 1A and 1B**, see **S1 Table** for all summary statistics), supporting a non-competitive interaction with *P. falciparum* as previously reported for virgin *An. gambiae* [17]. Of note, the number of eggs developed by mated

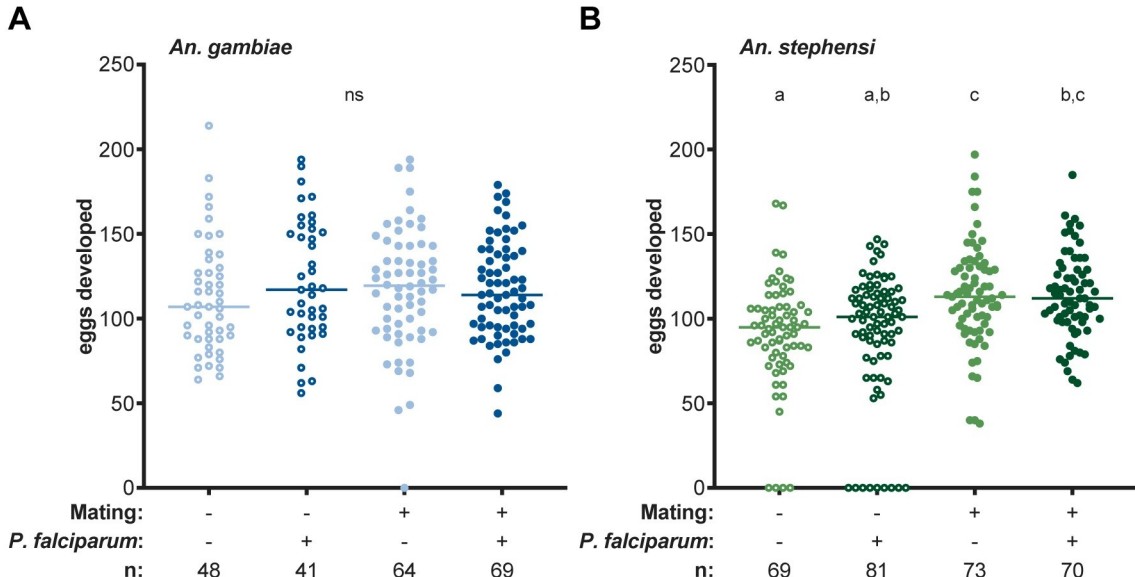

**Fig 1. *P. falciparum* infection does not affect the fecundity of *An. gambiae* and *An. stephensi* females.** *P. falciparum* infection does not affect the number of eggs developed by virgin and mated **(A)** *An. gambiae* females (infection: likelihood ratio test (LRT) $X^2_1 = 0.76$, $p = 0.38$; mating*infection: LRT $X^2_1 = 3.1$, $p = 0.08$) or **(B)** *An. stephensi* females (infection: LRT $X^2_1 = 0.007$, $p = 0.94$; mating*infection: LRT $X^2_1 = 0.99$, $p = 0.32$). In *An. stephensi*, mating increased the likelihood of egg development (LRT $X^2_1 = 16.26$, $p < 0.001$) and the number of eggs developed (mating: LRT $X^2_1 = 20.22$, $p < 0.001$). The data from this graph are from three replicates for each *Anopheles* species. Pairwise comparisons (letters) are computed via estimated marginal means. Lines represent medians, and n represents sample size. Open circles represent virgin females, closed circles represent mated females, and light and dark colors correspond to uninfected and infected females, respectively. ns = not statistically significant.

*An. gambiae* females was similar to virgins, as reported in some studies [30] but not in others [28] (**Fig 1A**, **S1 Table**). In *An. stephensi*, mating not only increased the likelihood of egg development but also boosted the number of eggs developed (**Fig 1B**, **S1 Table**), consistent with previous studies [29,30].

## Mating does not affect laboratory *P. falciparum* infections in *An. gambiae* and *An. stephensi*

In a subset of the virgin and mated *An. gambiae* and *An. stephensi* females used in the fecundity experiments detailed above, parasite intensity and prevalence of infection were determined by counting midgut oocysts 7 days pIBF. Interestingly, we detected no effects of mating on *P. falciparum* infection in *An. gambiae* (**Fig 2A**), as the same proportion of females became infected (infection prevalence, pie charts), and females developed the same number of oocysts (infection intensity) regardless of mating status (**S1 Table**). Similar results were obtained using *An. stephensi*, where again we observed no difference in prevalence and intensity of infection linked to mating (**Fig 2B**, **S1 Table**). In our hands, therefore, mating has no effect on *P. falciparum* infection in two different *Anopheles* species, in contrast to a recent study reporting that mating increases the susceptibility of *Anopheles coluzzii* females, a sibling species of *An. gambiae*, to *P. falciparum* [34].

## The mating-induced gene *MISO* mediates mosquito tolerance to *P. falciparum* in field infections

Our previous studies in virgin *An. gambiae* had shown that the steroid hormone 20E produced by females after blood feeding regulates the positive interaction between parasite and egg

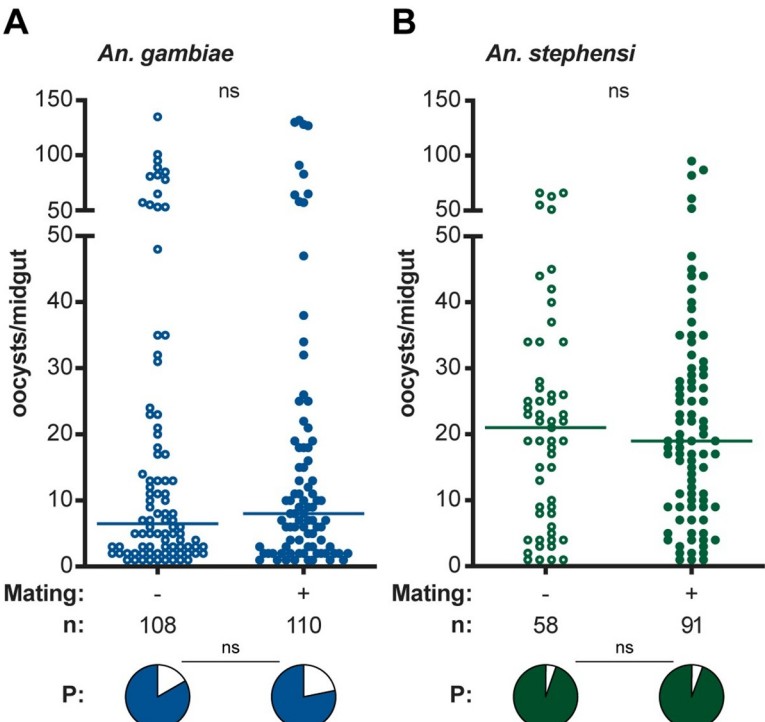

**Fig 2. Mating status does not impact *P. falciparum* laboratory infections in *An. gambiae* and *An. stephensi*.** Mating status does not affect prevalence (P, indicated with pie charts below the graph) or intensity of infection in (**A**) *An. gambiae* females (prevalence: LRT $X^2_1 = 1.08$, $p = 0.30$; intensity: LRT $X^2_1 = 0$, $p = 1.00$) or (**B**) *An. stephensi* females (prevalence: LRT $X^2_1 = 0.007$, $p = 0.93$; intensity: LRT $X^2_1 = 1.47$, $p = 0.23$). The data from this graph are from three replicates for each *Anopheles* species. Pairwise comparisons are computed via estimated marginal means. Lines represent medians, and n represents all data points including females that failed to develop oocysts. Females that developed no oocysts are omitted from the dot plots to represent intensity of infection. Open circles represent virgin females, closed circles represent mated females. ns = not statistically significant.

numbers [17]. Therefore, we set out to investigate whether mating-induced processes regulated by sexual transfer of the same steroid hormone 20E might be critical to the relationship between oogenesis and *P. falciparum* development in mated females. To this aim, we performed infections after silencing *MISO*, a female reproductive gene potently induced by male 20E in the female atrium after mating and that forms a complex with this steroid hormone, influencing timely lipid accumulation during oogenesis [28]. We reasoned that depleting MISO would be an effective strategy to partially disrupt male 20E-induced processes important for egg development and determine their impact during the course of *P. falciparum* infection. To gain a comprehensive picture of natural *Anopheles-Plasmodium* associations, infections were carried out in Burkina Faso using *P. falciparum*-infected blood collected from 5 donors with a range of gametocytemia values (**S2 Table**) and female mosquitoes from an *An. coluzzii* strain recently colonized from this region. After *MISO* depletion via RNAi injections of dsRNA targeting *MISO* (ds*MISO*), we allowed these females (or ds*LacZ*-injected controls, ds*Control*) to mate prior to being fed a *P. falciparum*-infected blood meal. We then deprived them of the opportunity to lay eggs (females retain eggs in these conditions, see Materials and Methods) in order to collect paired egg-oocyst data for each individual after dissecting females 7 days pIBF. The occurrence of mating was confirmed by detecting the presence of sperm in the spermatheca at the time of dissection.

When we analyzed oogenesis across different gametocytemia values, we found that gametocytemia alone had no effect on the number of eggs developed by *An. coluzzii* females

(**S1 Table**). Egg development in control females appeared to be more dependent on the random blood donor rather than on gametocytemia (**Fig 3A**). In contrast, we found that the interaction of gene silencing and gametocytemia significantly affected the number of eggs developed (**S1 Table**), whereby ds*MISO* females produced significantly fewer eggs at high gametocytemia infections relative to lower gametocytemia infections (**Fig 3A**). These data suggest that highly infectious *P. falciparum* blood meals reduce the reproductive fitness of females when this 20E-induced reproductive gene is depleted.

In both ds*MISO* and control groups, high gametocytemia in the donor blood resulted in higher oocyst loads, while infection prevalence was close to 70% or greater in all conditions and was not affected by gametocytemia (**Fig 3B**, **S1** and **S2 Tables**). Interestingly, no differences in either infection prevalence or intensity were observed in *MISO*-silenced versus control females, demonstrating that this reproductive gene does not impact parasite development (**Fig 3B**, **S1 Table**).

Additionally, when we examined the relationship between egg and oocyst numbers across all *An. coluzzii* infections by using paired egg-oocyst data, we found that in ds*MISO*-injected females egg numbers decreased with increasing oocyst loads, while they were not affected by infection in ds*Control* females (**Fig 3C**, **S1 Table**). Consistent with these findings, when analyzing the probability of egg development across the observed range of oocyst intensities, we found that while control individuals had a greater chance of developing eggs as the oocyst burden increased (**Fig 3D**), ds*MISO* females had a markedly reduced probability of egg development at increasingly high parasite loads (**Fig 3D**; **S1 Table**) and were 2.3% less likely to develop eggs for every additional oocyst they developed (Unit odds ratio = 0.977).

Combined, these data reveal that in the face of high intensity *P. falciparum* infections, MISO helps preserve reproductive fitness—as assessed by female fecundity—without affecting parasite survival, thereby acting as a key factor mediating *Anopheles* tolerance to *P. falciparum*.

## Discussion

In this study, we demonstrate that *P. falciparum* infections do not affect the fecundity of two important anopheline vectors—*An. gambiae* and *An. stephensi*—regardless of their mating status. Moreover, in *An. coluzzii* infections using natural *P. falciparum* isolates, we found that the number of eggs developed was not impaired by infections with increasing intensity, confirming previous data suggesting that *P. falciparum* has adopted a non-competitive evolutionary strategy in its natural mosquito vectors. Additionally, we found that females that have a higher likelihood of developing eggs harbor increasing oocyst loads, which expand on our previous findings in virgin *An. gambiae* where we found that egg and parasite numbers are positively correlated [17]. It is worth noting that past studies that showed costs of infection imposed by *Plasmodium* spp. were performed using rodent malaria parasites that are not naturally transmitted by anthropophilic *Anopheles* species [11–16]. The positive relationship between egg development and parasite numbers in this study and in our previous work [17] may be explained by the theory that organisms must balance resources between somatic, reproductive, and immune processes [35], whereby *An. coluzzii* females that devote more resources to egg development may have a suppressed immune response, enabling higher *P. falciparum* oocyst loads. Alternatively, processes triggered by blood feeding and regulating egg development post-mating may affect mosquito physiology in ways that also affect the probability of parasite survival in the midgut, while not being directly aimed at the parasite. A caveat of our study is that we only assessed the effects of parasite infection on the number of eggs developed after a first blood meal. Future studies will also be needed to address whether parasites may affect other components of mosquito fitness, including survival, fertility, and egg development over multiple gonotrophic cycles.

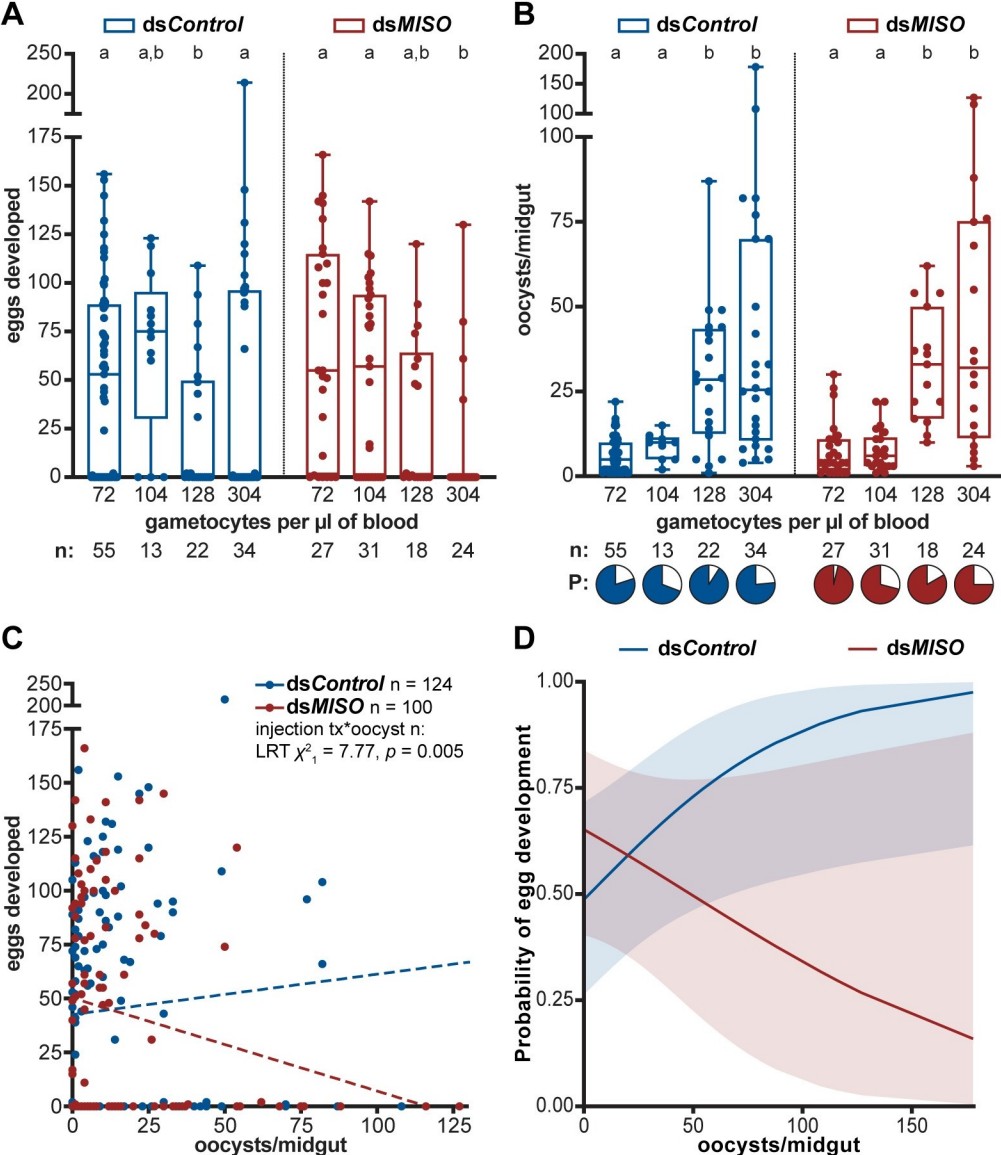

**Fig 3. MISO enforces tolerance to *P. falciparum* in field infections with gametocyte carriers. (A)** The number of eggs developed by *An. coluzzii* females silenced for *MISO* (ds*MISO*) is negatively impacted by *P. falciparum* infections with high gametocytemias, while this effect is not observed in ds*Control* mosquitoes (treatment*gametocytemia: LRT $X^2_3$ = 16.15, $p$ = 0.001). Four gametocytemias were tested (see **S2 Table**), with each dot representing paired egg-gametocytemia data for one female. **(B)** Both ds*Control* and ds*MISO* females harbor more oocysts when they are fed with higher gametocytemia blood meals (gametocytemia: LRT $X^2_3$ = 20.36, $p$ < 0.001), with no impact of *MISO* silencing (treatment: LRT $X^2_1$ = 0.85, $p$ = 0.36; treatment*gametocytemia: LRT $X^2_3$ = 0.85, $p$ = 0.84). *P. falciparum* prevalence (P, pie charts) is not affected by gametocytemia or *MISO* silencing (gametocytemia: LRT $X^2_3$ = 5.78, $p$ = 0.12; treatment: LRT $X^2_1$ = 0.60, $p$ = 0.44; treatment*gametocytemia: LRT $X^2_3$ = 4.63, $p$ = 0.20). Each dot represents paired oocyst-gametocytemia data for one female. For both **A** and **B**, data are represented as box and whisker plots with minimum and maximum values. Pairwise comparisons (letters) are computed via estimated marginal means. In **B**, females that developed no oocysts are omitted from the dot plots to represent intensity of infection. **(C)** *MISO* silencing breaks the interaction between egg and oocyst development, with higher oocyst intensities leading to fewer numbers of eggs developed (treatment*oocyst number: LRT $X^2_1$ = 7.77, $p$ = 0.005). Each dot represents paired egg-oocyst data for one female. One data point for ds*Control* (178, 88) is excluded from the graph to simplify data visualization. Dotted regression lines represent data trends only. tx = treatment, n = number, * = interaction. **(D)** While in control females the probability of egg development increases as oocyst intensity increases, *MISO*-silenced females have a lower likelihood of egg development as infection burden increases (treatment*oocyst number: LRT $X^2_1$ = 7.18, $p$ = 0.007). Curves represent the probability of egg development given *x* number of oocysts, with shading indicating 95% confidence intervals. In all panels, n represents sample size. All panels in this figure come from data collected from three independent replicates.

Moreover, we show that mating in *An. gambiae* and *An. stephensi* does not affect the intensity or the prevalence of *P. falciparum* infection. These data are in contrast to a recent report showing that mated *An. coluzzii* females harbor increased parasite burdens relative to virgins [34]. While we cannot explain the reasons behind this discrepancy, our findings were reproducibly obtained using two different *Anopheles* species.

The observation of no costs to fecundity induced by infection might be explained by considering that the interactions of *P. falciparum* and its ancestral lineages with *Cellia* species have been significant on an evolutionary scale. Indeed, human *Plasmodia* originated from ape populations in regions of Africa where these anophelines have thrived for millions of years (reviewed in [9]). Co-adaptations between mosquitoes and parasites may have arisen due to this longstanding association, favoring the evolution of tolerance mechanisms in the mosquito to limit costs of infections and of immune avoidance strategies in the parasite to avoid the induction of host resistance. A powerful example of the latter is provided by the evolution of the parasite gene *Pfs47* which mediates *P. falciparum* immune evasion in the mosquito [36]. This gene has undergone selection due to pressures imposed by the mosquito immune system in different *Anopheles* species, leading to distinct, geographically structured haplotypes [36–38]. Consistent with this scenario of mosquito-parasite co-evolution, we found evidence of a tolerance-mediating factor in *An. coluzzii*—one of the most important vectors of *P. falciparum* in sub-Saharan Africa. Specifically, in field studies we determined that the female protein MISO helps maintain mosquito fitness in the face of infection. Reproductive costs inflicted by parasites when MISO was depleted increased with higher infection intensities, while this depletion did not affect the parasite itself. Tolerance is a defense strategy whereby an organism reduces the fitness costs associated with an infection without impacting pathogen burden [31–33], and as such our results identify MISO as critical to *An. coluzzii* tolerance to *P. falciparum*. *MISO* is strongly induced in the female atrium by sexual transfer of 20E, which among mosquitoes has specifically evolved in the *Cellia* subgenus, with African species transferring higher titers of this steroid hormone [29]. Intriguingly, the evolution of *MISO* appears to have been driven by male 20E transfer, as MISO orthologues in species with the highest levels of male 20E show a greater degree of similarity than expected based solely on phylogenetic relationships [29]. When combined, our data suggest that the evolution of *MISO* may have promoted *Anopheles* tolerance to *P. falciparum*, thus contributing to the non-competitive strategy between these two organisms [17]. Tolerance strategies have previously been described in insects, including *Aedes* mosquitoes that transmit dengue and chikungunya [39]; however, to our knowledge this is the first demonstration of tolerance in an *Anopheles* vector of human malaria.

How could MISO affect female reproductive fitness during infection? Our previous work has shown that MISO ensures timely accumulation of lipids in developing eggs [28]. This protein forms a complex with male 20E [28], most likely in the atrium and ampullae, the tissues which connect the atrium to the ovaries. It is possible, then, that this complex may ensure the local release of sexually transferred 20E to preserve egg development in the face of midgut stress signals exacerbated by high infection loads. This model is compatible with known roles of steroid hormones in limiting damage induced by, for instance, oxidative stress [40–42]. Alternatively, the MISO-20E interaction may permanently 'prime' the ovaries to be competent for the correct accumulation of nutrients in mated females after blood feeding, a process known to be regulated by synthesis of the same hormone 20E in the female fat body. More recently, we have demonstrated that when blood meal-derived resources accumulate in the midgut rather than being delivered to the ovaries, as for instance when we impair egg development by silencing the 20E nuclear receptor EcR/USP, parasites are capable of utilizing those resources for growth [17]. It is therefore possible that when *MISO* is silenced, mosquito

resources that cannot readily be taken up by the ovaries may instead be utilized by parasites in some way, leading to a decrease in egg numbers that becomes more significant with higher parasite loads. Other explanations, including competition for resources due to sperm preservation in mated females, cannot be ruled out, and studying the cascades of events induced by *MISO*'s expression will help define the exact mechanism mediating *Anopheles* tolerance to *P. falciparum* infection. Combined with the finding that *MISO* is highly induced by mating in *An. coluzzii* females collected from natural mating swarms in the same villages in Burkina Faso [43], our results reveal that this female protein has an important function for maintaining reproductive fitness of field malaria vectors during *P. falciparum* infection.

## Materials and methods

### Ethics statement

Procedures involving human subjects were conducted in accordance with protocols approved by the Institutional Ethics Committee of the Centre Muraz (A003-2012/CE-CM), Bobo-Dioulasso, Burkina Faso and the Ethics Committee of the General Administration Department of the Ministry of Health of Burkina Faso (2014–0040). Volunteers with gametocytemias above 70 gametocytes/µl of blood were enrolled for the study following parental consent. *Anopheles* female mosquitoes at the Institut de Recherche en Sciences de la Santé (IRSS)/Centre Muraz, Bobo-Dioulasso, Burkina Faso were fed on rabbit blood in accordance with protocols approved by the Office of Laboratory Animal Welfare of the U.S. Public Health Service (Assurance Number: A5928-01) and the national committee of Burkina Faso (IRB registration #00004738 and FWA 00007038). Rabbits were cared for by veterinarians and trained personnel.

### Rearing of *Anopheles* mosquitoes

The G3 strain of *Anopheles gambiae* (composed of the two closely related sibling species *An. gambiae s.s.* and *Anopheles coluzzii*, herein referred to as *An. gambiae*) and the SDA-500 strain of *Anopheles stephensi* were used in laboratory experiments at the Harvard T. H. Chan School of Public Health (HSPH). The identity of the G3 and SDA-500 strain was confirmed by PCR and DNA sequencing of resulting amplified product [44]. For experiments conducted at the IRSS in Burkina Faso, *An. coluzzii* mosquitoes were used from a colony established in 2008 from wild-caught gravid females collected in the vicinity of Bobo-Dioulasso, repeatedly replenished with F1 from wild-caught mosquito females collected in Kou Valley (11˚23'14"N, 4˚24'42"W), 30 km from Bobo-Dioulasso [45,46], and identified by routine PCR-RFLP [44]. All mosquitoes were reared under standard conditions (26–28˚C, 65–80% relative humidity, 12-hour:12-hour light:darkness photoperiod). Adults from laboratory mosquito strains were fed on 10% glucose solution supplied *ad libitum*, and females were fed on human blood weekly (Research Blood Components, Boston, MA). Adults from the IRSS mosquito strain were fed on 5% glucose solution supplied *ad libitum*, and females were fed on rabbit blood weekly. For laboratory and field experiments, to ensure virgin status of mosquitoes prior to mating experiments, male and female mosquitoes were separated as pupae.

### Culturing *Plasmodium falciparum* NF54 parasites

The NF54 strain was confirmed as *P. falciparum* by PCR and DNA sequencing of amplified product [47]. Cultures were maintained at 37˚C in an incubator gassed to 5% oxygen, 5% carbon dioxide, 90% nitrogen. Asexual cultures of NF54 were maintained below 2% parasitemia at 5% hematocrit (O+ red blood cells, RBC) (Interstate Blood Bank) in complete medium composed of RPMI 1640 with L-glutamine and 25 mM HEPES, 10 µg/ml hypoxanthine, 0.2%

sodium bicarbonate, and 10% heat-inactivated O+ human serum (Interstate Blood Bank) according to established protocols [48]. To induce gametocytogenesis, cultures were grown to parasitemias of 3–10%, then split to 2% parasitemia in complete media at 5% hematocrit [49]. Parasites were cultured without subsequent splitting for 14–20 days to generate stage IV and V gametocytes, changing the media daily.

## Collection of natural isolates of *Plasmodium falciparum* parasites

Surveys for *P. falciparum* gametocyte carriers were conducted in the villages surrounding Bobo-Dioulasso, Burkina Faso. Blood smears were collected from pinprick samples from children ages 5–13, stained with 10% Giemsa, and assessed for *P. falciparum* infection levels by microscopy. The number of asexual and sexual parasites present per 1000 white blood cells was counted, and translated to gametocytes per μl of blood using the conversion 8000 white blood cells/μl of blood.

## Mating and fecundity assays in *Anopheles* mosquitoes

**In laboratory assays.** For laboratory experiments with *An. gambiae* and *An. stephensi*, matings were performed via two techniques: 1) 4 day-old virgin females were progressively added to cages containing 4–5 day-old males at a 1:2 ratio. Couples were then captured *in copula* as they dropped to the bottom of the cage using modified 50 mL centrifuge tubes [50]. 2) Females were mated using the forced mating technique, where the abdomen of anesthetized females are brought into contact with the abdomens of beheaded males to stimulate a mating response (https://www.beiresources.org). Females that failed to mate were screened out from the experiment by checking the abdomens of anesthetized females on a stereomicroscope Leica M80 with Fluorescence Light Source EL 6000 for autofluorescence resulting from the presence of a mating plug in the female atrium. All virgin females were separated from males as pupae and not subsequently exposed to males. Females were fed on *P. falciparum*-infectious or non-infectious blood at 1 day (d) post-mating (pM). 48 hours h pIBF, females were collected and stored in 70% ethanol at 4˚C. This timepoint was selected because starting at 3 d pIBF, females lay their eggs on the floor of the cage despite being deprived of a proper oviposition site due to the humidity in the custom-built glove box utilized in these experiments (Inert Technology, Amesbury, MA; see detailed infection protocol below). Of note, this experimental set-up prevents us from collecting paired oocyst intensity and egg number in the same mated female. Fecundity was assessed by dissecting the ovaries of these females in 1X phosphate-buffered saline (1X PBS) and manually counting eggs developed on a Nikon SMZ1000 stereomicroscope. A subset of females was kept to 7 d pIBF to assess *P. falciparum* infection intensity and prevalence.

**In field assays.** Two days post-eclosion, *An. coluzzii* females were added to cages of 3–4-day-old males in a 1:2 ratio. Following exposure to males over two nights, females were fed on *P. falciparum*-infected blood and transferred to new cages without males. Females were subsequently deprived of oviposition sites, and 7 d pIBF, females were collected and dissected to count the number of eggs developed and the number of oocysts in the midgut. Mating status was determined by assessing the presence/absence of sperm in the spermatheca. Cages were assessed regularly throughout the experiment to verify that females did not oviposit their eggs over the 7 day incubation period; no eggs were detected in the cages.

## Infections of *Anopheles* mosquitoes with *P. falciparum*

**Infections with *P. falciparum* NF54.** Cages of mosquitoes were fed on gametocyte cultures spun down and diluted in fresh RBC and human serum (Interstate Blood Bank) in a

custom-built glove box (Inert Technology, Amesbury, MA). To heat-inactivate gametocytes as non-infectious controls, aliquots of resuspended cultures were incubated at 42°C shaking at 500 rpm for 15 minutes [51]. 160 μl of these infectious and non-infectious culture samples were loaded into custom-designed glass membrane feeders (Chemglass Life Sciences) sealed with parafilm and heated to 37°C with a Haake D1 Immersion Circulator. Mosquitoes were allowed to feed for 30–60 minutes, adding more culture as necessary. All non- and partially-fed females were removed, and fully blood-fed mosquitoes were fed with 10% glucose solution *ad libitum* until dissection.

**Infections with natural *P. falciparum* isolates.** 2–4 ml of venous blood was drawn from gametocyte carriers, spun down, and serum was removed and replaced with an equal volume of *Plasmodium*-naïve AB serum. The sample was resuspended and loaded into custom-designed glass membrane feeders sealed with parafilm and heated to 37°C with a Julabo ED Immersion Circulator. Mosquitoes in cups were allowed to feed for 60–90 minutes. All non- and partially-fed females were removed from cages following the infection, and fully-fed mosquitoes were supplied 5% glucose solution *ad libitum* until dissection.

**Oocyst counts.** 7 d pIBF, female mosquitoes were collected in ethanol and transferred to 1X PBS. Midguts were dissected in 1X PBS, stained in 2 mg/ml mercury dibromofluorescein disodium salt (mercurochrome) (Sigma-Aldrich, St. Louis, MO) for 15 minutes, then loaded onto slides for visualization and manual counting at 200X using an Olympus Inverted CKX41 microscope.

## Gene expression knock-down using dsRNA

**Generating dsRNA from plasmids.** The plasmids utilized in these experiments were constructed for previous publications: pLL10-MISO (containing a 397 bp fragment of *MISO*, *AGAP002620*) [28] and pLL10-LacZ (containing a 816 bp fragment of *LacZ*) [28,52]. Plasmids were harvested from NEB Turbo chemically-competent *E. coli* cells (New England Biolabs, Ipswich, MA) and verified by DNA sequencing. Using primers containing the T7 sequence and specific to the plasmid backbone, PCR reactions were performed on plasmid DNA to amplify the coding sequence and generate knock-down constructs specific to each plasmid. The primers used to amplify the *MISO* and *LacZ* cassettes were pLL10-FWD: 5'-<u>TAATACGACTCAC TATAGGG</u>CTCGAGGTCGACGGTATCG-3' and pLL10-REV: 5'-<u>TAATACGACTCACTA TAGGG</u>CGGCCGCTCTAGAACTAG-3' (T7 regions underlined). PCR product sizes were confirmed via gel electrophoresis. dsRNA was then generated by incubating each PCR product with T7 enzyme using the MEGAscript T7 Transcription Kit (Thermo Fisher Scientific), purifying the dsRNA product by phenol-chloroform extraction, and re-suspending purified dsRNA in molecular grade de-ionized water (dH$_2$O) at a concentration of 10 μg/μl.

**dsRNA injections.** 1-day-old females were randomly assigned to injection groups (ds*MISO* or ds*Control*), anesthetized, and injected with 69 nl of 10 μg/μl dsRNA using a Drummond Scientific Nanoject II or III. Injected females were then mated and blood-fed on *P. falciparum*-infected blood as described above.

## RNA extraction, cDNA synthesis, and qRT-PCR

To assess silencing efficiency in *MISO* RNAi experiments, 8–12 atria were dissected from ds*MISO* and control females in 1XPBS approximately 24 h pM, pooled in TRIzol Reagent (Thermo Fisher Scientific), and stored at –80°C. Samples were homogenized with a hand-held homogenizer on ice, and RNA was extracted in TRIzol Reagent using 1-bromo-3-chloropropane, precipitated in isopropanol, and washed in 75% ethanol. RNA was quantified using a Nanodrop 2000c, and equal quantities of atrial RNA from *MISO* knock-down and control

groups were treated with TURBO DNase. The DNase-treated RNA samples were then loaded into 100 µl cDNA synthesis reactions with reagents at the following final concentrations: 1X First Strand Buffer, 5 mM DTT, 500 µM dNTP mix, 0.4 units RNaseOUT recombinant ribonuclease inhibitor, 2.5 µM random hexamers, and 1.25 units M-MLV reverse transcriptase (all reagents from Thermo Fisher Scientific). Primer sets (Integrated DNA Technologies) suitable for qRT-PCR were selected from a previous publication [28] where they were designed using Primer-BLAST (NCBI) [53] to limit off-target binding and to span exon-exon junctions when possible. The primer sequences were:

qRT-PCR MISO-F: AGACGATGGAGGGACTGATG
qRT-PCR MISO-R: GGATTCGCTTTCGTGCTG
qRT-PCR RPL19-F: CCAACTCGCGACAAAACATTC
qRT-PCR RPL19-R: ACCGGCTTCTTGATGATCAGA

Quantification was done using SYBR Green Detection, mixing 7.5 µl 1X Fast SYBR Green Master Mix (Thermo Fisher Scientific), 5 µl of a 1:10 dilution of cDNA, and primers at optimal concentrations, in duplicate for each sample on a StepOnePlus Real-Time PCR System (Thermo Fisher Scientific). Relative quantification of gene expression was calculated using the $\Delta C_T$ method, with the ribosomal protein *RpL19* (*AGAP004422*) used as the housekeeping gene. Samples were excluded from the analysis if the presence of >1 melt curve peak was detected or if amplification occurred in the non-template controls. Mean *MISO* silencing efficiency in *dsMISO*-injected females was 61% across 4 experiments.

## Quantification and statistical analysis

All statistical analyses were performed in R (version 4.0.2) [54]. A summary of all model statistics can be found in **S1 Table**. We used the glmmTMB package [55] to build Generalized Mixed Linear Models (GLMMs). For each of the models, we chose the most appropriate distribution through Akaike information criteria (AIC) comparisons. GLMMs with Gaussian errors were used to test the effects of mating, infection, and their interaction on the number of developing eggs in the ovaries of *An. gambiae* (**Fig 1A**) and *An. stephensi* (**Fig 1B**). A GLMM with binomial errors was used to test the effect of mating, infection, and their interaction on the likelihood of developing eggs (egg prevalence) in *An. stephensi* (**Fig 1B**). Similar binomial GLMMs were used to test the effect of mating on oocyst prevalence in *An. gambiae* (**Fig 2A**) and *An. stephensi* (**Fig 2B**). GLMMs with zero-truncated negative binomial errors were used to test the effect of mating on oocyst intensity (the number of oocysts in infected mosquito females) in both *An. gambiae* (**Fig 2A**) and *An. stephensi* (**Fig 2B**). For each of these GLMMs, experimental replicate was set as a random effect. The effects of gametocytemia, *MISO* silencing (referred to as "treatment") and their interaction on egg number in *An. coluzzii* (**Fig 3A**) was analyzed using a GLMM with zero-inflated negative binomial errors. The effects of gametocytemia, treatment, and their interaction on oocyst prevalence and intensity in *An. coluzzii* were analyzed using a binomial GLMM and a zero-truncated negative binomial GLMM, respectively (**Fig 3B**). The effects of oocyst intensity, treatment, and their interaction on egg number and prevalence in *An. coluzzii* were analyzed using a zero-inflated negative binomial GLMM (**Fig 3C**) and a binomial GLMM (**Fig 3D**), respectively. In these GLMMs, because the experiment on *An. coluzzii* was conducted over 3 replicates and using a total of 5 gametocyte carriers (one replicate with one carrier and two replicates with two carriers each), carrier (parasite isolate) was nested within replicate and considered together as nested random effects. Model simplification used stepwise removal of terms, followed by likelihood ratio tests (LRT). Term removals that significantly reduced explanatory power ($p < 0.05$) were retained in the minimal adequate model. All pairwise comparisons were done using the "emmeans" function

in the emmeans R package that computes estimated marginal means and automatically corrects for unequal sample sizes using the Tukey-Kramer method [56].

## Supporting information

**S1 Table. Model statistics related to analyses in Figs 1–3.** Generalized Linear Mixed Models (GLMMs) were constructed and analyzed in R. The test/model used in each figure and for each comparison is outlined with individual likelihood ratio test (LRT) outputs. Significant tests ($p < 0.05$) are bolded.
(PDF)

**S2 Table. Parameters for experimental infections of *An. coluzzii* with field isolates of *P. falciparum*.** Samples of *P. falciparum*-infected blood were collected from 5 gametocyte donors and fed to *An. coluzzii* mosquitoes silenced for *MISO* or a control gene. The number of gametocytes for each sample was counted per 1000 white blood cells on blood smears, then converted to the number of gametocytes per μl of blood assuming a standard white blood cell count of 8000 cells per μl of blood. The outcome of infection was determined by counting the number of oocysts developed per female 7 d pIBF.
(PDF)

## Acknowledgments

We wish to thank Kate Thornburg, Emily Lund, and David Clarke for help with mosquito rearing and insectary procedures. We also wish to thank Naresh Singh, Jamaica Siwak, and Ping Lui for assistance with *P. falciparum* culture and infections in the laboratory at the Harvard T. H. Chan School of Public Health, and Bali-Jean Bazié and Raymond Hien for assistance with *P. falciparum* infections and mosquito dissections at the Institut de Recherche en Sciences de la Santé in Bobo-Dioulasso, Burkina Faso. We are grateful to Daniel E. Neafsey (Harvard T.H. Chan School of Public Health, Boston, MA, USA) for helpful discussions and feedback on this manuscript. We would also like to thank Carolina Barillas-Mury and Alvaro-Molina Cruz (National Institutes of Health, Bethesda, MD, USA) for guidance and resource sharing for *P. falciparum* NF54 culturing.

## Author Contributions

**Conceptualization:** Perrine Marcenac, Flaminia Catteruccia.

**Formal analysis:** Perrine Marcenac, Adam South, Thierry Lefèvre.

**Funding acquisition:** Perrine Marcenac, Flaminia Catteruccia.

**Investigation:** Perrine Marcenac, W. Robert Shaw, Evdoxia G. Kakani, Sara N. Mitchell, Eryney Marrogi, Daniel G. Abernathy, Rakiswendé Serge Yerbanga, Abdoulaye Diabaté, Thierry Lefèvre.

**Methodology:** Perrine Marcenac, Flaminia Catteruccia.

**Resources:** Roch K. Dabiré.

**Supervision:** Thierry Lefèvre, Flaminia Catteruccia.

**Visualization:** Perrine Marcenac, Thierry Lefèvre.

**Writing – original draft:** Perrine Marcenac, Flaminia Catteruccia.

**Writing – review & editing:** Perrine Marcenac, W. Robert Shaw, Evdoxia G. Kakani, Sara N. Mitchell, Kristine Werling, Thierry Lefèvre, Flaminia Catteruccia.

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
