## [Decision Letter · Decision Letter 0]

12 Oct 2020

Dear Dr Catteruccia,

Thank you very much for submitting your manuscript "A mating-induced reproductive gene promotes Anopheles tolerance to Plasmodium falciparum infection" for consideration at PLOS Pathogens. As with all papers reviewed by the journal, your manuscript was reviewed by members of the editorial board and by several independent reviewers. The reviewers appreciated the attention to an important topic. Based on the reviews, we are likely to accept this manuscript for publication, providing that you modify the manuscript according to the review recommendations.

Sincerely,

Kenneth D Vernick

Associate Editor

PLOS Pathogens

Kirk Deitsch

Section Editor

PLOS Pathogens

Kasturi Haldar

Editor-in-Chief

PLOS Pathogens

orcid.org/0000-0001-5065-158X

Michael Malim

Editor-in-Chief

PLOS Pathogens

orcid.org/0000-0002-7699-2064

Reviewer Comments (if any, and for reference):

Reviewer's Responses to Questions

**Part I - Summary**

Reviewer #1: This study examined whether P. falciparum interferes with egg development in mated An. gambiae and An. stephensi females. The study builds up on earlier work from the same group (Werling et al. 2019) that found a positive correlation between egg numbers and parasite loads in virgin females. Although this result suggested a non-competitive interaction between P. falciparum and An. gambiae, it remained to be determined whether this finding extended to reproductively active, mated females and other Anopheles species. The fitness cost of malaria parasite infection in mosquitoes is a relevant and longstanding question in vector biology.

Overall, I found this manuscript to be compelling and well written, and I think it will be of great interest to both vector biologists and a more general audience of evolutionary biologists. I have a few comments about the interpretation and discussion of the results, but with these relatively minor considerations taken into account, I would not hesitate to recommend this manuscript for publication.

Reviewer #2: In this manuscript Marcenac et al. provide new information on reproduction and tolerance to Plasmodium infection. Through the use of reproduction/mating bioassays coupled with functional and infection experiments the authors provide convincing evidence that the MISO gene is involved in mosquito tolerance to Plasmodium infection, allowing it to maintain reproductive fitness.

The manuscript adds much needed new knowledge to understand the effects of Plasmodium infection on mosquito reproductive fitness and provides some potential mechanistic evidence for the absence of detrimental effects of Pf infection in the most important Anopheline vectors. The manuscript is easy to read and well organized when presenting the data via graphs or during the discussion section. I commend the authors for providing alternative hypothesis, not shying away from writing other potential explanations of the observable data.

**Part II – Major Issues: Key Experiments Required for Acceptance**

Reviewer #1: (No Response)

Reviewer #2: I don't see any major experiment that needs to be conducted. The experiments appear solid.

**Part III – Minor Issues: Editorial and Data Presentation Modifications**

Reviewer #1: 1) My most significant criticism is about the primary phenotype examined to measure the fitness cost of P. falciparum infection. The number of developed eggs within the ovaries after a single blood meal is only a partial measure of the mosquito’s reproductive output. Ideally, not only the total number of eggs laid across multiple gonotrophic cycles throughout a female’s lifetime (fecundity) should have been considered, but also the viability of these eggs (fertility) should have been assessed to more accurately estimate the overall fitness. I suggest the authors tone down their claims about fitness and the fitness cost of infection, and better contextualize their findings. For example, the statement that “P. falciparum infections do not affect the reproductive fitness” (line 197) should explicitly mention the caveat that the phenotype examined (number of developed eggs within the ovaries) did not fully encapsulate fitness.

Following up on the above point, why were the females not allowed to lay their eggs in the experiments? Oocyst numbers could have been determined after oviposition to obtain paired fecundity and parasite load measurements. Interfering with the natural process and time frame of oviposition may have altered life-history and resource allocation trade-offs. The authors may also consider justifying/discussing this point.

2) Another aspect that may deserve to be discussed further relates to the concept of host tolerance. Tolerance is often represented as the negative slope of host fitness as a function of parasite burden. Individuals with a shallower slope are more tolerant because they maintain higher fitness at high parasite loads. In the present study, the relationship between the probability of egg development and the number of oocysts was negative in MISO-depleted individuals, consistent with a relatively lack of tolerance, however the same relationship had a positive slope in the control mosquitoes (Fig. 3D). Could this positive slope reflect the cost of resistance of individuals with low parasite numbers, if females who are able to clear the infection are less likely to produce eggs ? Or to the contrary, a positive effect of high parasite loads on the probability of egg development?

In the same line, the previous work from this group (Werling et al. 2019) found a positive relationship between egg and parasite numbers in laboratory assays. Because this specific result was not confirmed in the field-based experiments of the present study (Fig. 3C), it would be interesting to discuss the potential reasons underlying this discrepancy.

3) The statistical analyses were performed according to very high standards, however it would have been useful to specify how the error distribution was chosen in the various models. Also, why were oocyst prevalence and intensity analyzed by separate GLMMs, whereas zero and non-zero egg counts were analyzed together?

Reviewer #2: It would be good if the authors describe how many independent experiments were conducted and represented in each graph. Also, I am curious to know how the authors decided to employ Tukey HSD to do pairwise analysis. One of the assumptions for using Tukey HSD is that of equal sample sizes for each treatment; but on the graphs (in particular 2B) has vastly different sample sizes. Perhaps I am misunderstanding something here?

PLOS authors have the option to publish the peer review history of their article (what does this mean?). If published, this will include your full peer review and any attached files.

Reviewer #1: **Yes: **Louis Lambrechts

Reviewer #2: No
---

## [Editor Report · Decision Letter 1]

11 Nov 2020

Dear Dr Catteruccia,

We are pleased to inform you that your manuscript 'A mating-induced reproductive gene promotes Anopheles tolerance to Plasmodium falciparum infection' has been provisionally accepted for publication in PLOS Pathogens.

Best regards,

Kenneth D Vernick

Associate Editor

PLOS Pathogens

Kirk Deitsch

Section Editor

PLOS Pathogens

Kasturi Haldar

Editor-in-Chief

PLOS Pathogens

orcid.org/0000-0001-5065-158X

Michael Malim

Editor-in-Chief

PLOS Pathogens

orcid.org/0000-0002-7699-2064
---

## [Editor Report · Acceptance letter]

16 Dec 2020

Dear Dr Catteruccia,

We are delighted to inform you that your manuscript, "A mating-induced reproductive gene promotes *Anopheles* tolerance to *Plasmodium falciparum* infection," has been formally accepted for publication in PLOS Pathogens.

Best regards,

Kasturi Haldar

Editor-in-Chief

PLOS Pathogens

orcid.org/0000-0001-5065-158X

Michael Malim

Editor-in-Chief

PLOS Pathogens

orcid.org/0000-0002-7699-2064